# Obesity, Bariatric Surgery and Obstructive Sleep Apnea—A Narrative Literature Review

**DOI:** 10.3390/medicina59071266

**Published:** 2023-07-07

**Authors:** Krzysztof Wyszomirski, Maciej Walędziak, Anna Różańska-Walędziak

**Affiliations:** 1Department of Human Physiology and Pathophysiology, Faculty of Medicine, Collegium Medicum, Cardinal Stefan Wyszynski University in Warsaw, 01-938 Warsaw, Poland; k.wyszomirski@uksw.edu.pl (K.W.); aniaroza@tlen.pl (A.R.-W.); 2Department of General, Oncological, Metabolic and Thoracic Surgery, Military Institute of Medicine—National Research Institute, Szaserów 128 St., 04-141 Warsaw, Poland

**Keywords:** bariatric surgery, obstructive sleep apnea, obesity surgery, sleep-related hypoventilation, apnea-hypopnea index

## Abstract

The purpose of this review was to analyze the available literature on the subject of obesity and obstructive sleep apnea. We searched for available articles for the time period from 2013 to 2023. Obesity is listed as one of the most important health issues. Complications of obesity, with obstructive sleep apnea (OSA) listed among them, are common problems in clinical practice. Obesity is a well-recognized risk factor for OSA, but OSA itself may contribute to worsening obesity. Bariatric surgery is a treatment of choice for severely obese patients, especially with present complications, and remains the only causative treatment for patients with OSA. Though improvement in OSA control in patients after bariatric surgery is well-established knowledge, the complete resolution of OSA is achieved in less than half of them. The determination of subpopulations of patients in whom bariatric surgery would be especially advantageous is an important issue of OSA management. Increasing the potential of non-invasive strategies in obesity treatment requires studies that assess the efficacy and safety of combined methods.

## 1. Introduction

The World Health Organization lists obesity as one of the most important health issues in the modern world. It is estimated that more than half of the world’s population will be overweight or obese by 2030 [1]. Excess body weight also affects children—6% of girls and 8% of boys aged 5–19 years were overweight or obese in 2016 [2]. What is to be emphasized is that the problem of obesity is a challenge both for the developed and developing world [3].

Obesity itself is an independent risk factor for numerous conditions [4]. Referring to sleep disorder characteristics, excess body mass has a well-established position as a major, independent risk factor of OSA [5,6,7]. Among the numerous risk factors for OSA, body mass index—BMI ≥ 25 kg/m^2^ plays a core role, while a prospective cohort study estimated that 10% weight gain could lead to a 32% increase in the apnea-hypopnea index (AHI), whereas analogical weight loss is associated with a 26% fall of AHI [8,9]. BMI is directly correlated with the severity of OSA syndrome [10]. The feature distinguishing obesity from other risk factors for OSA is its modifiability, making excess body weight clinically the most important risk factor for OSA [11].

OSA itself can contribute to weight gain in obese patients in a mechanism associated with other OSA complications: sleep fragmentation, sympathetic activation, insulin resistance, and others [5]. Additionally, obesity is associated with a higher risk of nocturia in OSA patients, a symptom that increases alongside OSA severity [12] (Reviewer 1).

The OSA diagnosis is based on clinical symptoms evaluated in questionnaires and is confirmed by polysomnography. The frequency of restriction-related events during sleep can be assessed with the apnea-hypopnea index (AHI) or respiratory disturbance index (RDI). The essential difference between AHI and RDI is that respiratory-related arousals are taken into account in RDI. OSA severity is defined by polysomnography and AHI/RDI determination. Mild OSA is defined as AHI/RDI ≥ 5 but <15. AHI/RDI > 15 but ≤30 corresponds with moderate OSA, whereas AHI/RDI > 30 indicates severe OSA [3]. Assessing the impact of medical intervention in OSA can be based on questionnaires but preferably on polysomnography [13].

The estimated prevalence of OSA in the general population is 4% in males and 2% in females, making OSA the most common sleep-breathing disorder [11,14]. However, the American Association of Sleep Medicine (AASM) evaluates that the real prevalence of OSA is much higher. The prevalence of OSA in the severely obese population is higher than in the general population, and there are estimated to be about 60 to 80% of patients awaiting bariatric surgery [15].

## 2. Material and Methods

This is a review article with the purpose of analyzing the available literature on the subject of obstructive sleep apnea and other sleep disturbances in bariatric patients. The “National Library of Medicine PUBMed” was comprehensively searched for the following terms: “obstructive sleep apnea, bariatric surgery”, “obstructive sleep apnea”; obesity surgery”, “sleep-related hypoventilation”, “apnea-hypopnea index” by two independent researchers. We searched for available original articles on the subject of obesity and sleep disorders, including a time period from 2013 to 2023. Original studies assessing sleep disturbances, mostly OSA in patients prior, and post-bariatric surgery, were included under the supervision of the senior researcher. We have divided our manuscript into separate chapters, including OSA, Obesity Hypoventilation Syndrome (OHS), the effect of bariatric surgery, and the conservative treatment of OSA.

### 2.1. Obstructive Sleep Apnea (OSA) in Obese Patients

OSA, primarily defined as an anatomical abnormality concomitant with obesity, now has a well-established position as a manifestation of a metabolic syndrome with a rather molecular than strictly anatomical mechanism of development [14]. Insulin resistance and an increased leptin level are independent of obesity associated with OSA. One of the considered mechanisms of leptin’s impact on OSA is its direct influence on the respiratory center in the central nervous system [3]. OSA and obesity coexist with elevated levels of proinflammatory cytokines. Obesity facilitates low-grade inflammation in many complex molecular pathways. Remittent hypoxia episodes, which are a fundamental pathology caused by OSA, also favor inflammation. Obesity and OSA may aggravate each other by the exacerbation of an inflammatory state [16]. Importantly, in the context of bariatric surgery, Tirado et al. proved a statistically highly significant (*p* < 0.01) decrease in inflammation-associated molecules (CRP, HO-1, IL-6, IL-1β, TNF-α) after the procedure with no statistically significant differences between the Roux-en-Y gastric bypass (RYGB) and sleeve gastrectomy [16]. Regarding how respiratory drive leptin is recognized as an essential stimulant of ventilation, increased leptin levels in obese patients reflect leptin resistance, with a decreased respiratory drive as one presentation of central leptin resistance [17,18]. These listed facts are proof of the molecular origin of OSA [12,14]. There are numerous risk factors independently associated with an increased likelihood of OSA, including male sex, BMI > 35 kg/m^2^, smoking, gastroesophageal reflux disease (GERD), hypothyroidism, acromegaly, and benzodiapine use [19].

The necessity of effective treatment for OSA arises from severe, recognized complications of OSA, including poor control of hypertension, other cardiovascular comorbidities, and premature mortality [20,21]. OSA is an important risk factor for the most common arrhythmia—atrial fibrillation. Evidence strongly suggests that appropriate OSA management can improve atrial fibrillation control [22]. Positive airway pressure (PAP) in different modes, preferably continuous positive airway pressure (CPAP), is a gold standard for OSA treatment with proven effectiveness in reducing OSA symptoms, complications and improving the quality of sleep [7,23]. Although CPAP is the “golden method” for OSA treatment, there are other methods in OSA therapy. Mandibular advancement devices (MADs) are anti-snoring mouthpieces designed to physically move the jaw forward and expand the airway. There are also some new methods, including Lin oral appliances or even transoral robotic surgery [24,25,26] (Reviewer 2). In the context of considered bariatric surgery, preoperative CPAP improves the safety of general anesthesia and the procedure itself [27].

Though a decrease in body mass is an advantageous strategy in patients with OSA, it leads rather to a reduction in symptoms than to the resolution of OSA [28]. Another common difficulty is the fact that no more than 10% of patients with obesity treated only with dietary or behavioral regimes maintain weight loss in the long-term perspective [9]. Due to the temporary effect of behavioral strategies, AASM proposes bariatric surgery as the leading adjunctive therapy for OSA in obese patients. The remission rate of OSA after bariatric surgery was estimated by AASM to be 40%. Sustainable weight loss seems to contribute to the success of invasive strategies.

### 2.2. Obesity Hypoventilation Syndrome (OHS)

OHS is characterized by its three elements: BMI ≥ 30 kg/m^2^, daytime hypercapnia defined as PaCO2 ≥ 45 mmHg, sleep breathing disturbances, and an absence of other comorbidities that may explain alveolar hypoventilation. The difference distinguishing OHS from OSA is the presence of daytime alteration. Daytime hypercapnia corresponds with chronic or acute respiratory failure type 2. Respiratory alterations that are present during the day could aggravate during sleep. The prevalence of OHS was estimated at 0.4% of the general population. OSA is a morbidity that coexists with OHS; obstructive sleep apnea is diagnosed in 90% of patients with OHS, whereas some studies have suggested that even 70% of patients burdened with OHS present a severe form of OSA defined as AHI ≥ 30 events/h) [17]. Daytime hypercapnia can be present in OSAS, but several OSAS patients may not have this status. In a study by Kawata et al., CPAP was found to reduce the rate of patients with severe OSAS presenting with daytime hypercapnia by 50% and was better tolerated by patients with a higher BMI [29] (Reviewer 1). A misdiagnosis in patients suffering from OHS is very common. According to Masa et al., 75% of patients with OHS were previously misdiagnosed with obstructive lung disease. CPAP is the treatment of choice in patients with OHS and severe OSA, whereas NIV is used preferably in patients with OHS and AHI < 30 [17]. The perioperative risk was elevated in patients with OHS. Preoperative PAP treatment seems to mitigate the risk associated with the procedure both in patients with OSA and OHS [30]. There are no data on the specific impact of bariatric surgery on OHS. Persistent weight loss after the procedure and the well-established positive effect of bariatric surgery on OSA control are facts supporting the presumable positive effect of bariatric surgery on OHS. 

### 2.3. OSA, CPAP and Weight Loss

As obesity and weight gain are related both to the increased occurrence and severity of OSA, and OSA is an independent risk factor of weight gain, there exists an adverse positive feedback loop. Weight gain increases the risk of OSA, and OSA leads to further weight gain. Therefore, it would be logical if CPAP therapy and successful OSA treatment led to weight loss. However, there are conflicting data, with the majority of available studies on the subject showing that CPAP therapy can be associated with weight gain. In a randomized controlled trial by Quan et al. in a group of 1105 participants with OSA, 6 months of CPAP treatment was associated with weight gain, and patients with the highest adherence to CPAP therapy had the highest weight gain [19]. These results were confirmed in a meta-analysis by Kovács et al., who stated that CPAP did not lead to weight loss over several years of therapy [31]. Similar conclusions were drawn by Drager et al. in their meta-analysis regarding the effects of CPAP on body weight in patients with OSA, including 3181 patients. CPAP therapy resulted in significant weight gain in obese and overweight patients; therefore, they suggested an additional treatment for obesity [32]. In a study by Garcia et al., CPAP was associated with an increase in insulin levels and insulin resistance, and 40% of patients experienced weight gain, even though their treatment was successful in reducing hypoxia [33]. A meta-analysis by Chen et al. showed that CPAP led to weight increase in patients with no cardiovascular disease at the baseline, whereas it was correlated with weight decrease in patients with cardiovascular disease at the baseline, with the opposite effect of dysglycemia at the baseline [34]. It should be emphasized that the adverse effect of CPAP-induced weight gain, even though of statistical significance, in most studies was not higher than 0.5 kg [35]. There are also studies that have presented the opposite effect of CPAP therapy on body weight in obese patients. In a study by Pocienė et al. in a group of 119 obese patients with a baseline BMI of 41 ± 8 kg/m^2^, significant weight loss was observed after 3 months in 34% of patients and after 9 months in 47% of patients, with no increase in body weight in 62% of patients after 3 months and 47% after 9 months [36]. Further research is needed to analyze the background for these different effects of CPAP therapy in selected subpopulations and to find an answer to whether CPAP-induced increases in body weight are due to an increase in body fat, lean body mass, or water compartments (Reviewer 1).

### 2.4. Effect of Bariatric Surgery on OSA Treatment in Obese Patients

In compliance with the current guidelines, bariatric surgery has been indicated in patients with a BMI > 40 independently of coexisting comorbidities or in patients with BMI > 35 with a history of comorbidities, including diabetes mellitus type 2, heart failure, hypertension, or OSA [11]. Though numerous studies have been conducted in recent years to define the impact of bariatric surgery on OSA, there remains a need to gather further information, especially in RCTs. 

Furlan et al. randomly assigned patients into groups that were treated with Usual Care (nutritional, psychological intervention, physical activity) or RYGB [9]. The follow-up lasted for 3 years. Compared to UC, RYGB was related to a significant improvement in numerous parameters, including BMI, EDS, and neck and waist circumference. Importantly, a median increase in AHI was observed in the UC group, while the RYGB group showed a significant decrease in AHI. Bariatric surgery gave an incomparably better chance to cure OSA than UC. The obtained differences between groups were statistically significant with *p* < 0.05. This study has its disadvantages which arise from a rather small group of patients (n = 24 patients treated with RYBG, n = 13 patients treated with UC). 

Nastalek et al. included obese patients who were diagnosed with OSA using screening tools [11]. In addition to sleep questionnaires, each patient was assessed with polysomnography preoperatively and 12 months after the procedure. In total, 44 patients diagnosed with OSA were treated with bariatric surgery (sleeve gastrectomy (31/44) or RYBG (13/44)), underwent follow-up hospitalization, and were included in the final analysis. After CPAP titration and continuation, a significant improvement in numerous parameters, including AHI, ODI, and mean SpO2, was observed. All these listed findings were of high statistical significance with *p* < 0.001. In total, 25 patients from 44 (56.8%) achieved the normalization of AHI. Though CPAP remained a gold standard to treat OSA, treatment with CPAP was burdened with a low compliance problem. In this study, CPAP was used over less than half of the days, which indicated low compliance in these patients. Compared to a preoperative assessment, post-operative polysomnography showed a significant improvement in numerous parameters, including the AHI (44.9 vs. 29.2), oxygen desaturation index (ODI) (43.6 vs. 18.3), mean hemoglobin saturation (93% vs. 95%) and snoring (21.6 vs. 4.5%). High statistical significance was obtained (*p* < 0.001). A moderate correlation between the percentage loss of excess AHI and the percentage loss of excess BMI was observed. This effect of bariatric surgery on OSA is consistent with the current literature [5,37,38] (Reviewer 1). Bariatric surgery resulted in an approximately low rate of normalization regarding sleeping disturbances (16%), while in 5% of cases, OSA severity increased. The most probable scenarios seem to be stabilization (39%) or a decrease in OSA severity without complete normalization (41%). 

Another study comparing the effects of bariatric surgery versus non-invasive treatment was Feigel-Guiller et al. [39]. Obese patients with OSA treated by non-invasive mechanical ventilation (NIV) were included in this study. The target was to assess the differences in NIV weaning in patients treated with gastric surgery or INC at 1, 3, and 10 years of follow-up. Patients (n = 70) were randomly allocated into two groups—the first group was treated with laparoscopic adjustable gastric banding (LAGB), and the second group was treated non-invasively by intensive nutritional care (INC). Data were collected at years 1, 3, and 10 after intervention. LAGB was associated with a significantly greater weight loss than INC, with 15% vs. 6% after the first year (*p* < 0.001) and 14% vs. 3% after the third year (*p* < 0.001). Despite greater weight loss and a positive trend in NIV weaning in patients treated with LAGB, the difference was not statistically significant both after the first and third years. The INC group showed a decrease in AHI in years 1 and 3; however, this result was not statistically significant. The decrease in AHI in the LAGB group was statistically significant after one and three years of follow-up. Data collected after ten years of follow-up showed that a high percentage of patients in both groups required another bariatric procedure. 

Auclair et al. evaluated the impact of biliopancreatic diversion with a duodenal switch (BPD-DS) on OSA and systemic hypertension [40]. In this study, a 77% remission rate in OSA was achieved in a twelve-month follow-up. The molecular profile of patients who achieved resolution of OSA was similar to those who did not, with the exception of a high-sensitivity C-reactive protein. The reduction in this parameter was statistically and significantly greater in patients who were cured of OSA than in patients who did not achieve a resolution. This study could suggest high the efficacy of BPD-DS in achieving an OSA resolution. The disadvantage of this study is a rather small probe: from the sixty-two patients included in this study, forty underwent a procedure. Twenty patients of twenty-six with OSA had their OSA resolved. As the effects of BPD-DS diverged from the results presented in other studies, this method requires following assessments in studies involving a bigger probe. 

While the general improvement in OSA control after bariatric surgery is well-established knowledge, there is an increasing need to conduct studies that compare different techniques. Nowadays, leading techniques include LRYGB and the more conservative laparoscopic sleeve gastrectomy (LSG). Wölnerhanssen et al. conducted a study based on merged Swiss and Finnish data [41]. The comparison of the results of LSG to LRYGB was conducted in all fields in relation to metabolic syndrome (MetS), including OSA. No statistically significant difference was noted between LSG and LRYGB regarding OSA’s remission rate. The OSA control was similar in both groups after five years of follow-up as well. This study should be followed by numerous works of research to assess different invasive techniques on their effectiveness and safety. 

The results of numerous studies assessing bariatric surgery’s impact on OSA control are relatively consistent: bariatric surgery decreases OSA symptoms and can lead to OSA resolution. Yet, there is an increasing need to conduct studies assessing certain subpopulations. The prevalence of OSA and obesity is also alarmingly high among adolescents [42]. Amin et al. conducted a study proving an early improvement in OSA symptoms control and laboratory findings in patients between fifteen and twenty years old [43]. The disadvantage of this study is the small probe of patients used. 

### 2.5. Conservative Treatments for OSA

A study assessing OSA control in the long term was presented by Kuna et al. [44] In this study, two conservative regimes were compared. Intensive lifestyle intervention (ILI) gave a greater chance of reducing OSA severity compared to diabetes support and education (DSE). The improvement in the control of OSA symptoms was correlated with body mass reduction, baseline AHI and factors independent of weight change. A higher probability of OSA remission after ten years of follow-up was observed in the ILI group (34.4% vs. 22.2%).

Positive and promising effects of both invasive and conservative strategies in OSA management signalize a need to conduct studies that compare the efficiency and safety of bariatric surgery to acknowledged conservative strategies, for example, ILI. Combined behavioral, pharmacological, and surgical strategies also require studies to assess their effectiveness [45]. Regarding pharmacological intervention pharmaceutics from flozins and the glucagone-like peptide 1 analog, groups should be considered that can offer supportive treatment to invasive procedures. Neeland et al. proved the beneficial effect of empagliflozin in patients with diabetes mellitus 2 and coexisting OSA [46].

The OSA remission rate may be increased by the avoidance of supine sleeping positions. Joosten et al. examined patients that did not meet OSA resolution criteria after significant weight loss [28]. In total, 22% of them had resolved their OSA after changing their daily sleep position to non-supine. 

### 2.6. Future Perspectives

OSA treatment is a complex problem as the mainstay form of therapy—CPAP—even though effective, is associated with lowering through time patients’ compliance. Additionally, CPAP is correlated with small but statistically significant weight gain, and weight gain is a well-known risk factor for OSA. The reduction in body weight that may be acquired through the means of bariatric surgery can eliminate one of the most important risk factors for the occurrence and severity of OSA; therefore, bariatric surgery should be included in the guidelines for OSA treatment. It is still to be established whether treatments should start with CPAP and bariatric surgery should be added to augment the positive treatment effect of CPAP or whether treatment should start with bariatric surgery and CPAP should be added after achieving weight loss (Reviewer 1).

## 3. Conclusions

OSA is inseparably associated with obesity; therefore, its prevalence is much higher in the obese population than in the general population. The effect is acquired both in the mechanism associated with weight loss and in a weight loss-independent pathway. According to current guidelines, the complete resolution of OSA is expected in approximately 40% of patients undergoing bariatric surgery. The remission of OSA after bariatric surgery is not necessarily in correlation with excessive weight loss (EWL%), as some studies present that even though EWL% is good, its effect on the remission of OSA symptoms is not satisfactory. On the contrary, the results of other studies have shown a high impact of weight loss after bariatric surgery on OSA symptoms. The exact time-point of introducing bariatric surgery in OSA treatment is still to be established, including whether the treatment should be started with bariatric surgery and followed by introducing CPAP after preliminary weight loss, or whether CPAP should be introduced in the postoperative period, or perhaps if CPAP should be introduced before bariatric surgery as it might ameliorate the intubation conditions of anesthesia. (Reviewer 2) The remission rate can be ameliorated by the addition of previously established OSA treatment.

There remains a need to develop new strategies for treatment combining bariatric surgery with non-invasive methods and to assess them in randomized control trials. 

## Data Availability

Not applicable.

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
