# Peer review of "Obesity, Bariatric Surgery and Obstructive Sleep Apnea—A Narrative Literature Review"

_medicina, 2023, doi:10.3390/medicina59071266_

Round 1

Reviewer 1 Report

Authors should be congratulated for their work.

The manuscript is well-written and easily readable. The topic addressed is challenging. Indeed, OSAS represents a health bothering concern that underlies several symptoms and diseases, often related to other conditions.

- A check of the information reported in the paragraph "Obesity hypoventilation syndrome" must be required. The daytime hypercapnia, indeed, could be present in the OSAS, but several OSAS patients could not have this status (PMID: 18079218).

- Are data available on the role of CPAP in lowering the weight of OSAS patients? 

- The reference should be implemented to improve the scientific soundness of the work. I must suggest this novel systematic review on OSAS and nocturia that Authors could read to catch several aspects of OSAS physiopathology and the relation between BMI, obesity, and the disease itself (PMID: 37167825). Moreover, in line 165 several references are required to confirm the Authors' theory. 

- Finally, a paragraph on future perspectives must be added to make the work comprehensive. 

Author Response

Dear Reviewer 1,

Thank you very much for your review.

According to your suggestion, we corrected the "Obesity hypoventilation syndrome" and added information that the daytime hypercapnia is not present in several OSAS patients, with an adequate reference to literature.

Following your remark, we added a short paragraph about relation between OSAS, CPAP and body weight loss. The majority of available data shows that CPAP results in weight gain, small, but statistically significant, however there are also studies presenting the opposite.

We implemented the reference of the indicated new systematic review about nocturia and OSAS patients, adding information about the problem of OSAS, BMI and nocturia.

We added references to the line 165, and we repositioned the mentioned sentence as by our mistake it was put in the wrong position in the paragraph.

As you suggested, we added a paragraph on future perspectives.

Reviewer 2 Report

Topic: obesity and obstructive sleep apnea—a narrative literature review

Q1: The topic seems like a report of a student for the “narrative”.

Bariatric surgery might seem to be included in the topic.

How is this quality for this paper? Why not try organize a systemic review?

---- ----

Introduction:

---- ---

Materials and methods

Q2:Flow chart is good for the reader more clear.

How many studies were surveyed?

Considering bias and conflicts of interest among the included studies? Any systematic error? Performing assessments of risks of bias?

--- ---

Results

Page 3 line 94: ..treatment for OSA…

Q3: there are some methods for OSA. CPAP is the“golden method”. Mandibular advancement devices (MADs) were anti-snoring mouthpieces designed to physically move the jaw forward and expand the airway. The Lin oral appliances is a new method for OSA.

Q4: Why not try to draw the forest plot?

--- ----

Q5: There are lots of papers for the OSA. Why the authors selected only these few papers. What are the evidence levels of these papers?

--- ----

Page 5 line 211: conservative treatments for OSA

Q6: Bariatric surgery effect, it is better to compare the effect. Is this the major aim for this paper? The authors why not selected more studies and compare the effect.

--- ----

Page 5 line 231

Conclusion

Beneficial effect of bariatric surgery on OSA control is well-established knowledge.

Q7: if this idea is “well- established”. What is the key value of this paper?

Author Response

Dear Reviewer 2,

Thank you for your review.

Q1: Indeed, your deduction was true – the main author was still a student at the beginning of work on this review. That is also why the construction of the manuscript was decided to be a narrative, not a systematic review.

Following your suggestion, we included bariatric surgery in the topic.

Materials and methods

Q2: Thank you for this remark, however the flow chart was not possible to be created, as our research was limited to analysis of present knowledge about obstructive sleep apnea and obesity.

We analyzed 43 studies.

Authors of our manuscript did not participate in creation of any of cited papers, so we declared no conflicts of interest. Studies chosen for the review did not have systematic errors.

Results

Q3: According to your suggestion, we added information about methods of treatment other than CPAP.

Q4: Thank you for this suggestion, however due to the research conception it was not possible to create the forest plot.

Q5: We chose the papers that seemed  the most relevant on the subject. The evidence levels of the papers mentioned were from I to IV.

Q6:The main purpose of the study was to analyze the impact of bariatric surgery on OSA, we added the paragraph about conservative therapy only to indicate other possible methods of treatment.

Conclusion

Q7: Thank you for this remark. We agree that you are right and the indicated sentence is mistaking, therefore we removed it from the manuscript as the idea is not exactly established and the time-point of introducing bariatric surgery in OSA treatment is to be subject to further studies.

Round 2

Reviewer 1 Report

Good work.

Reviewer 2 Report

several questions did not replied by the authors.